# Improved Unsupervised Stitching Algorithm for Multiple Environments SuperUDIS

**DOI:** 10.3390/s24165352

**Published:** 2024-08-19

**Authors:** Haoze Wu, Chun Bao, Qun Hao, Jie Cao, Li Zhang

**Affiliations:** 1Instrument Science and Technology, Beijing Institute of Technology, Beijing 100081, China; 3120220652@bit.edu.cn (H.W.); baochun@bit.edu.cn (C.B.); qhao@bit.edu.cn (Q.H.); zhanglsky@126.com (L.Z.); 2School of Optoelectronic Engineering, Changchun University of Science and Technology, Changchun 130013, China; 3Yangtze Delta Region Academy, Beijing Institute of Technology, Jiaxing 314003, China

**Keywords:** image stitching, deep learning, unsupervised stitching, chroma balance

## Abstract

Large field-of-view images are increasingly used in various environments today, and image stitching technology can make up for the limited field of view caused by hardware design. However, previous methods are constrained in various environments. In this paper, we propose a method that combines the powerful feature extraction capabilities of the Superpoint algorithm and the exact feature matching capabilities of the Lightglue algorithm with the image fusion algorithm of Unsupervised Deep Image Stitching (UDIS). Our proposed method effectively improves the situation where the linear structure is distorted and the resolution is low in the stitching results of the UDIS algorithm. On this basis, we make up for the shortcomings of the UDIS fusion algorithm. For stitching fractures of UDIS in some complex situations, we optimize the loss function of UDIS. We use a second-order differential Laplacian operator to replace the difference in the horizontal and vertical directions to emphasize the continuity of the structural edges during training. Combined with the above improvements, the Super Unsupervised Deep Image Stitching (SuperUDIS) algorithm is finally formed. SuperUDIS has better performance in both qualitative and quantitative evaluations compared to the UDIS algorithm, with the PSNR index increasing by 0.5 on average and the SSIM index increasing by 0.02 on average. Moreover, the proposed method is more robust in complex environments with large color differences or multi-linear structures.

## 1. Introduction

Nowadays, digital images have played an increasingly important role, whether as a convenient method of information transmission or as one of the important media for machines to perceive the outside world. With the development of technology, more and more application scenarios require equipment to cover a wider field of view. However, due to the limitations of the hardware performance of the acquisition equipment, the captured images often only cover a limited field of view. The existing methods of capturing large fields of view all have some defects, such as fisheye cameras, camera scanning, or camera arrays. Although the use of special equipment such as fisheye lenses provides a solution for large fields of view, the overall cost of this equipment is relatively high, and the lens has serious distortion. As shown in Figure 1, this kind of distortion cannot completely offset the distortion of the image even after post-image processing, eventually leading to local blurring and information loss. Using a single camera for scanning imaging reduces costs and expands the field of view. However, it cannot meet the real-time requirements due to the limitations of the camera frame rate and other factors. Using a camera array to obtain a larger field of view fits the real-time requirements, but each camera is independent of each other, which greatly reduces the efficiency of extracting information. In order to solve the above problems, we introduced image stitching technology.

Image stitching refers to the fusion of two or more pictures that may come from different times, different spaces, or different sensors based on similar parts on the premise of ensuring image resolution and splicing different images with limited fields of view into a large-scale image. Scenes in the field of view can obtain more comprehensive, more realistic, and more detailed information, making it easier and faster to process and understand information. This algorithm has applications in a variety of fields, such as autonomous driving, remote sensing measurement [1], emergency disaster relief [2], medical imaging, surveillance video, and virtual reality.

In the past few decades, image stitching methods have been roughly divided into traditional stitching methods and deep learning stitching methods. Traditional splicing methods can be roughly divided into two parts: image registration and seam splicing. Image registration mainly includes feature point extraction, feature point matching, and homography transformation, while seam splicing mainly includes finding seams and image fusion. At present, traditional image stitching methods cannot adapt to more scenes to achieve versatility due to their reliance on geometric structure and photometric differences, while deep learning stitching methods can be applied to larger scenes but cannot adapt to large parallax images.

To overcome the limitations of feature-based solutions and supervised deep solutions, Nie et al., propose an unsupervised image stitching method called UDIS [3]. The algorithm consists of two parts: unsupervised alignment and unsupervised fusion. In the alignment stage, a homography network based on ablation loss optimization is designed. In the fusion stage, a low-resolution deformation branch and a high-resolution refinement branch are designed in the reconstruction network to achieve high-resolution unsupervised image stitching. Afterwards, the author optimizes the network and implements UDIS [4] which can overcome large parallax scenes. The overall idea of the UDIS algorithm is novel. The unsupervised alignment part avoids the problems of inaccurate feature point capture and inaccurate matching in the image registration part. The unsupervised fusion part avoids the problem of traditional image fusion, which relies on image luminosity, and also solves the problem of deep learning. The fusion algorithm is difficult to deal with because of the large parallax.

However, the UDIS algorithm still has some flaws, which prevent it from giving good splicing results when facing some complex scenes. First, as shown in Figure 2a,d, the color transition at the edge of the seam is not natural enough. Second, as shown in Figure 2b,e, in some images the seams are not perfect, there are still some misalignment errors. Moreover, as shown in Figure 2c,f, due to the adaptive distortion of the grid, some overall structures are destroyed, especially linear structures. Additionally, the resolution of the results stitched by the UDIS algorithm is low, and the texture details are lost.

In the image distortion part, this article combines Superpoint, which can quickly and accurately extract feature points. And Lightglue can adapt to difficulty to ensure computing speed while avoiding the linear structure distortion and low result resolution of the UDIS unsupervised distortion algorithm. At the same time, a chroma balance algorithm is added before fusion, which corrects the problem of color mutation in the splicing result of the UDIS fusion algorithm when facing large color differences. In addition, this paper also uses the Laplacian operator to adjust the loss function of the UDIS fusion algorithm so that it can better fuse structural edges. Finally, Superpoint, Lightglue, the chroma balance algorithm, and the improved UDIS fusion algorithm are combined to become a more excellent SuperUDIS algorithm.

The main contributions of this paper can be summarized as the following three points:The proposed algorithm applied the Superpoint feature point and LightGlue feature matching methods to the image stitching algorithm at first. The effect of Superpoint combined with LightGlue is significantly better than the distortion part of the ORB and UDIS algorithms on the HPatches dataset.In the proposed algorithm, we exploit a deep learning-based method of UDIS for seam splicing and use the chroma balance algorithm to preprocess the input image of UDIS to solve the problem of obvious color faults. At the same time, the loss function is replaced by the Laplacian operator. The difference in direction optimizes the splicing result.The proposed algorithm combines Superpoint, Lightglue, and the UDIS fusion algorithm. The three algorithms complement each other. Superpoint and Lightglue enhance the precision of image registration and solve the problem that the UDIS algorithm has limited resolution and poor effects in specific environments. In addition, the utilization of floating-point masks in the UDIS algorithm contributes to smoother stitching outcomes.

## 2. Related Work

Image stitching algorithms can be categorized into traditional methods and deep learning-based methods. These methods can be divided into two steps: image warping and image fusion. Image registration mainly establishes the geometric correspondence between images so that they can be transformed into a common reference frame and then reprojected in the same reference coordinate system. Image fusion is mainly to remove the gap by changing the gray level near the boundary so as to achieve a smooth transition between images.

### 2.1. Traditional Stitching Algorithm

Image registration aims to align one image (to be stitched) with another image (reference image) by determining a spatial transformation for pairs of images within a dataset captured under various conditions, such as different acquisition equipment, different time, different shooting angles, etc. The points corresponding to the same position in space in two images can be correctly matched to achieve the purpose of information fusion.

Traditional image stitching methods typically detect key points or line segments. The classic methods include Moravec corner detection [5], Harris corner detection [6], SIFT [7,8], SURF [9,10], BRIEF [11], ORB [12,13,14], BRISK [15] and KAZE [16,17,18]. Then, we minimize projection errors by aligning these geometric features to estimate parametric warping.

Based on this process, other methods are further optimized for the existing problems. To eliminate dislocation parallax, the splicing model is extended from global homography to local homography [19]. In addition, there are some methods to help maintain the natural structural shape of non-overlapping areas. For instance, DFW [20], SPW [21], and LPC [22] leveraged line-related consistency to preserve geometric structures.

Although image registration can map multiple images to the same coordinate system, due to different parameters among multiple sensors, spatial shelter, and other factors, different brightness and pixel misalignment will inevitably occur after registration. Therefore, an image fusion algorithm is needed to smooth the transition of images and improve their naturalness.

Uyttendaele and Eden et al., first proposed the emergence fusion algorithm [23], which gradually mixed the pixel values of two images in the image fusion region. Then, a fusion algorithm based on the optimal suture line is proposed. By calculating the different energy values of the overlapping regions of the image and minimizing the energy value to an optimal suture line, the fusion of the two regions is realized [24,25]. In the subsequent discussion of the method, chroma [26], saturation [27], edge gradient [28], texture constraints [29], etc. [30,31], are gradually added to the minimum energy exchange function for optimization. Additionally, Daeho Lee et al., eliminated artifacts by detecting multiple sutures and selecting the best one among them, as well as by pyramid fusion [32,33,34,35].

### 2.2. Stitching Algorithm Based on Deep Learning

Yi et al., proposed a learning-based Invariant Feature Transform (LIFT) network architecture [36], which simulates SIFT by estimating key points, their orientation, and their rotation-invariant descriptors. DeTone et al., proposed a Superpoint model [37], which first trained a self-supervised network model called MagicPoint using basic graphics, then carried out random homologous transformation on this basis, and used the Superpoint model for end-to-end training to learn feature points and extract their descriptions. Finally, the prediction of feature points and descriptors is realized. SuperGlue, proposed by Paul-Edouard Sarlin et al., is a neural network that matches two sets of local features by jointly finding corresponding points and rejecting mismatching points [38]. Paul-Edouard Sarlin et al., making improvements to SuperGlue, proposed LightGlue, which is more accurate, easy to train in terms of memory and computation, and able to self-adapt to image matching difficulty to avoid wasting resources [39].

Wu et al., proposed to combine the advantages of GAN and gradient-based image mixing algorithms to develop a new framework called GP-GAN [40]. Zheng C et al., proposed Localin Reshuffle (LRNet) [41], which can maintain a smooth gradient domain of mixed images and transmit local texture and illumination in the process of stitching. Inspired by Laplace’s pyramid fusion method [42], Zhang et al., proposed a densely connected multi-stream fusion (MLF) network [43], which can effectively fuse foreground and background image information of different scales. Lu et al., proposed a new bidirectional content transfer module that simultaneously adopted a context attention mechanism and an adversarial learning scheme to ensure spatial and semantic consistency in the mixing process [44].

## 3. SuperUDIS

The architecture of our proposed SuperUDIS is shown in Figure 3.

SuperUDIS contains four key components: feature point capture, image alignment, chroma balance, and image fusion. The Superpoint algorithm serves as the primary tool for feature point capture, while the Lightglue algorithm is primarily employed for image alignment. Additionally, the chroma balance algorithm proposed in this paper is used for chroma balance, and the unsupervised fusion model of the UDIS algorithm trained by the loss function improved in this paper is used for image fusion. Superpoint, Lightglue, and the UDIS fusion algorithm are classical algorithms. Therefore, the basic processes and advantages of the above three algorithms are only briefly introduced in Section 1, Section 2 and Section 4, respectively. The chroma balance algorithm and the improved UDIS fusion algorithm are detailed in Section 3 and Section 5, respectively.

### 3.1. Superpoint

Since the pretrained model of Superpoint is trained by using basic geometric figures called MagicPoint, the model can have a good extraction effect for the corner feature points in basic geometric figures. Moreover, because the model has undergone homographic adaptive training (as shown in Figure 4), it can still extract the basic graphics after translation, rotation, scaling, or perspective transformation and has rotation invariance and scale invariance.

In addition, Superpoint utilizes a VGG-style encoder with the overall network structure to reduce the image dimension and then simultaneously performs decoding operations to generate feature point positions and feature descriptors, as shown in Figure 5, which greatly reduces the operation time. In the real world, various scenes are composed of translation, scaling, rotation, perspective transformation, and other methods of basic geometric figures. Thus, Superpoint can accurately and efficiently extract feature points in the real scene, laying a good foundation for subsequent work.

### 3.2. Lightglue

LightGlue is a structure with multiple identical layers stacked with the position p and feature descriptor d of two-dimensional points in the image as inputs. Each layer includes a self-attention unit and a cross-attention unit. In the process of calculation, the network will update the representation of each point, screen and remove the points that are impossible to match, and the classifier will decide whether to stop reasoning at each layer, thus minimizing the amount of computation. Finally, a lightweight header computes partial match results. The overall structure is shown in Figure 6.

Lightglue leverages sophisticated self-attention and cross-attention mechanisms to enhance the precision of feature point matching. Moreover, the model is endowed with the ability to predict the confidence of the predicted results, and the depth and width of the model are self-adaptive. If a simple match is encountered, the prediction can be finished in the early stages, saving the operation time. In the case of complex matching, it can go through more rounds of prediction to improve the prediction accuracy. In addition, when an unmatched point is encountered in the multi-layer reasoning of the model, the model will exclude it in advance to avoid redundant computations.

### 3.3. Chroma Balance Algorithm

Due to variations in time, sensors, and other factors, multiple images may exhibit differences in lighting and exposure. Fusing these images directly without any preprocessing, the final stitching result image may have a sudden change in color at the edge of the stitching seam, resulting in the final stitching image with obvious stitching seams. Therefore, we propose a chroma balance algorithm. The main idea of the chroma balance algorithm is to give each input image a gain coefficient so that the image intensity of the overlapping part is as equal or similar as possible. The overall process is shown in Figure 7.

Firstly, based on the matching feature points obtained by the Superpoint and Lightglue algorithms above, the relative position of the two images is initially determined and the masks generated. The mask multiplies with the corresponding image to get the pre-stitched image. Then we multiply the two masks to get the overlapping mask. The overlapping mask multiplies with the two pre-stitched images, respectively, to obtain the overlapping part of the two images.

Then we can define the loss function e according to the target, as shown in Equation (1), where $N_{ij}$ is the number of pixels in the overlapping area, $g_{i}$ and $g_{j}$ are the gain coefficient of the two images, respectively. $\bar{I_{i}}$ and $\bar{I_{j}}$ are the average intensity of the two images in the overlapping area, respectively. The specific calculation is shown in Equation (2), where $R(u_{i}),G(u_{i}),B(u_{i})$ are the intensity of the red, green, and blue components at a certain point in the overlapping area, respectively.
(1)$$e={\sum_{i=1}^{n}\sum_{j=1}^{n}N_{ij}[g_{i}\bar{I_{i}}-g_{j}\bar{I_{j}}]}^{2}$$
(2)$$\bar{I_{i}}=\frac{\sum_{u_{i}\in R(i,j)}\sqrt{R^{2}(u_{i})+G^{2}(u_{i})+B^{2}(u_{i})}}{N_{ij}}$$

After solving, we find that $g_{i}=g_{j}=0$ is always the optimal solution to Equation (1). But this is not the result we hope to get, so we need to make a correction to the above equation to ensure that when the gain coefficient is 0, the loss function is not the optimal solution. The correction result is shown in Equation (3).
(3)$$e=\sum_{i=1}^{n}\sum_{j=1}^{n}N_{ij}[\frac{{\left(g_{i}\bar{I_{i}}-g_{j}\bar{I_{j}}\right)}^{2}}{\sigma_{N}^{2}}+\frac{{\left(1-g_{i}\right)}^{2}}{\sigma_{g}^{2}}]$$
where $\sigma_{N}\;and\;\sigma_{g},$ respectively, represent the standard deviations of error and gain, $\sigma_{N}$ and $\sigma_{g}$ are assigned as 10 and 0.1. Take the derivative of Equation (3) and then make its derivative 0 to obtain a closed-form solution, as shown in Equation (4).
(4)$$\frac{\delta e}{\delta g_{i}}=2\left(\sum_{j=1,j\neq i}^{n}\frac{N_{ij}I_{ij}^{2}}{\sigma_{N}^{2}}+\sum_{j=1}^{n}\frac{N_{ij}}{\sigma_{g}^{2}}\right)g_{i}-2\sum_{j=1,j\neq i}^{n}\frac{N_{ij}I_{ij}I_{ji}g_{j}}{\sigma_{N}^{2}}-2\sum_{j=1}^{n}\frac{N_{ij}}{\sigma_{g}^{2}}$$

By the same token, take the partial derivative of each gain and set it to 0, so we can get a multivariate linear equation. The gain coefficient of each image can be obtained by solving the linear equations.

### 3.4. UDIS Fusion Algorithm

As shown in Figure 8, the fusion part of UDIS first takes the mask and input image as input and puts them into the U-Net-like network for stitching. However, this will lead to the mixing of semantic features between different images and affect the judgment of the network. To solve this problem, UDIS first extracts semantic features with shared weights using the encoder of the network, then subtracts the features of the target image from the reference image, replaces them with residuals at each resolution in the decoder, and finally gets the output mask of the two images.

The loss function is divided into two parts, and the specific calculation is expressed as Equation (5). They are boundary term $L_{boundary}^{c}$ and smooth term $L_{smoothness}^{c}$ respectively, and the smooth term is divided into difference smooth term $l_{D}$ and stitching smooth term $l_{S}$.
(5)$$L^{c}=\alpha L_{boundary}^{c}+\beta L_{smoothness}^{c}$$

The boundary term is mainly used to encourage the end of the stitching joint to be the intersection point of the distorted image boundary, and the specific calculation method is shown in Equation (6).
(6)$$L_{boundary}^{c}={\left\|(S-I_{wr})\cdot M_{br}\right\|}_{1}+{\left\|(S-I_{wr})\cdot M_{bt}\right\|}_{1}$$

We use S to represent the overall spliced image, $M_{br}$ and $M_{bt}$ represent the boundary of the overlapping part of the mask, and finally, the front and back terms, respectively, represent the boundary part of the intersection of the two images. This loss limits the boundary pixel of the overlapping area in S to be from one of the boundaries of the two graphs and fixes the end point of the seam at the boundary crossing point of the overlapping area as much as possible.

The smoothness term is divided into the difference smoothness term $l_{D}$ and the stitching smoothness term $l_{S}$. The former describes the chroma difference on the difference image, and the latter describes the continuity of the seam edge of the stitching image. In the differential smooth term, we adopt the simplest photometric difference as $D={\left(I_{wr}-I_{wt}\right)}^{2}$, the differential smooth term is defined as Equation (7); the concatenated smooth term is calculated using the first-order difference in the direction, defined as Equation (8).
(7)$$l_{D}=\sum_{i,j}\left|M_{cr}^{i,j}-M_{cr}^{i+1,j}\right|\left(D^{i,j}+D^{i+1,j}\right)+\sum_{i,j}\left|M_{cr}^{i,j}-M_{cr}^{i,j+1}\right|\left(D^{i,j}+D^{i,j+1}\right)$$
(8)$$l_{S}=\sum_{i,j}\left|M_{cr}^{i,j}-M_{cr}^{i+1,j}\right|\left(S^{i,j}-S^{i+1,j}\right)+\sum_{i,j}\left|M_{cr}^{i,j}-M_{cr}^{i,j+1}\right|\left(S^{i,j}-S^{i,j+1}\right)$$

In addition, for a learning system, the back-propagation of the gradient can be affected by using a prediction mask with strict integers, and in addition, strict integer masks tend to produce discontinuous content in the results. So the algorithm sets the mask to a float. After the optimized floating-point mask is obtained, the two learning masks and the input image are dotted and added to generate the final stitching image result.

### 3.5. Loss Function Optimization

Figure 9 shows several groups of images obtained by using the UDIS algorithm. It can be seen that the box of the resulting picture has some poor effects, such as dislocation and blurring. Unsupervised stitching can be divided into three parts in the loss function, which are the boundary term, the stitching smooth term, and the difference smooth term. The boundary term is mainly used to encourage the two ends of the joint to be in the same part of the mask boundary. The stitching smoothness term optimizes the smoothness of the stitching results based on the continuity of overlapping parts. The differential smoothness term optimizes the smoothness of the image chroma. The problems of misalignment and ambiguity indicate that the stitching smoothness terms need to be optimized.

The stitching smoothness term is mainly used to constrain the continuity of the seam edge of the image after stitching. In the loss function of the UDIS fusion algorithm, the stitching smoothness term calculates the difference between horizontally and vertically adjacent pixels, respectively. Then the difference results of the horizontal and vertical adjacent pixels of the mask are multiplied and added to find the average, and the average value is finally taken as the loss function. Figure 10a,d is used as the original figure, and Figure 10b,e is the transverse difference graph that has been processed in the same way as the loss function of the UDIS algorithm. The dislocation in the stitching image is often the most intuitive at the edge of the structure, and it also has the most direct impact on the image perception. Therefore, we think that we should pay more attention to the edge of the structure when calculating the concatenated smooth term, and the smoothness of the rest of the parts outside the edge is constrained by the differential smooth term. As shown in Figure 10b,e, the transverse difference in the loss function of UDIS can indeed highlight the edge of the structure. However, due to the simple deviation of other pixels, there are also results close to the edge outside the edge of the structure, which leads to the weakening of the edge of the structure in the different results.

As shown in Equation (9), the difference in the loss function of the previous UDIS algorithm can be viewed as the first-order differential of a two-dimensional image in the *X*-axis direction or the *Y*-axis direction. The first-order differential (partial derivative) reflects the speed of gray change, and many times only the change of gray level (first-order differential), but it cannot prove that there is edge or structure information, and the second-order differential just makes up for this defect.
(9)$$\frac{\delta f}{\delta x}=f\left(x+1,y\right)-f\left(x,y\right)$$

In order to strengthen the continuity of the edge of the structure, the first-order difference in direction is proposed to be replaced with the Laplacian convolution. Then we multiply the difference in the result of the mask and finally get the improved loss function.

Like the Sobel operator, the Laplacian operator is a common edge extraction operator in image processing and belongs to the spatial sharpening filter operation. The Laplacian operator is a second-order differential operator in n-dimensional Euclidean space, defined as the divergence of the gradient. The Laplace operator is a second-order differential linear operator. In image edge processing, the first-order differential can reflect the speed of gray change, while the second-order differential can reflect the intensity of gray change rate, that is, the sudden change of gray. Therefore, the second-order differential has stronger edge localization ability and a better sharpening effect. In image edge processing, we choose to directly use the second-order differential operator instead of the first-order differential. As for Figure 10c,f, the image processed by the Laplacian operator will not produce the same interference points as the first-order difference but only leave some structural boundaries, which greatly enhances the proportion of structural edges in the loss function, thus making the overall structure of the final stitching result more continuous.

The specific calculation equation of the Laplacian operator is shown in Equation (10), so its matrix expression is shown in Figure 11a. It can be seen that the method gets the same result in the directions of 90° up and down, left and right, but the result is different in the direction of 45°. Therefore, we extend the Laplace operator so that it is also non-directional in the direction of 45°. The matrix representation of the extended Laplace operator is shown in Figure 11b.
(10)$$\left\{\begin{array}{c} \nabla^{2}f(x,y)=\frac{\delta^{2}f}{\delta x^{2}}+\frac{\delta^{2}f}{\delta y^{2}}\\ \frac{\delta^{2}f}{\delta x^{2}}=f(x+1,y)+f(x-1,y)-2f(x,y)\\ \frac{\delta^{2}f}{\delta y^{2}}=f(x,y+1)+f(x,y-1)-2f(x,y) \end{array}\right.$$

Figure 11c,d are based on the results obtained by operators Figure 11a,b and convolution of the target image, respectively. It can be seen that the results obtained by the extended Laplacian operator are obviously better than the basic Laplacian operator, and its edge extraction effect is more obvious, which is more conducive to the model’s retention of structural edges.

## 4. Experiment and Result

### 4.1. Dataset and Details

**Dataset**: For the image registration phase, we mainly used the classical dataset HPatches for the experiment. For the image fusion stage, we mainly use the SEAGULL dataset [45] and the LPC dataset [22] for experiments.

**Details**: For the unsupervised stitching part of the improved UDIS algorithm, we set the sum to 10,000 and 2000, respectively, and the number of iterations to 20 rounds. The experimental model was trained using a single GPU, NVIDIA GeForce RTX 2080 Ti from Santa Clara, CA, USA, and the experimental results were tested using a single GPU, NVIDIA RTX GeForce RTX 3060 Ti from USA.

### 4.2. Comparison to the Previous Methods

#### 4.2.1. Image Registration Experiment

We compare our approach to the unsupervised distortion part of the ORB + RANSAC, and UDIS methods. Three methods were used to process the images of HPatches [46], respectively, and then the pair of feature points were obtained by the three methods for camera parameter estimation to obtain the H matrix. The H matrix obtained by the three methods was compared with the true value error of the H matrix in the dataset of HPatches.

HPatches can be divided into two parts, consisting of illumination and viewpoint, respectively. Among them, the illumination group is taken under the same view point but different illumination conditions (from light to dark), and the H matrix between the same groups in the illumination group should be 0, as shown in the upper two lines in Figure 12. The viewpoint group is taken from different viewpoints under the same illumination conditions, and the H matrix of the viewpoint group within the same group should be different, as shown in the lower two lines in Figure 12.

In this paper, the H matrix calculated by the three methods is averaged after each element of the ground truth of HPatches is different. Due to the large value, the average value of each H matrix is calculated to obtain the final data. The final data are shown in Figure 13.

It can be seen from the figure above that among the three methods, ORB + RANSAC and UDIS algorithms have high errors in H-matrix estimation, while Superpoint + Lightglue used in this paper has the smallest errors in H-matrix estimation. The smaller the error is, the closer the homography matrix obtained after feature extraction and image matching is to the actual position relationship between the two, and the closer the two are to the true-value image after restoration.

Superpoint can accurately extract effective feature points in images, while Lightglue can match the extracted feature points accurately and quickly to achieve registration. The combination of these two excellent algorithms ensures that the fusion image can be matched accurately, even if there are only a few overlapping parts. Accurate registration lays a good foundation for the subsequent fusion.

#### 4.2.2. Image Fusion Experiment

We compare the results of the proposed method qualitatively and quantitatively with previous methods, including AutoStitch and the UDIS algorithm. It should be noted that, in both datasets, some images failed to be stitched using the AutoStitch algorithm. The following results are the average values calculated after removing the failure results.

**Quantitative**: In the quantitative comparison, we mainly compare the peak signal-to-noise ratio (PSNR) and Structural SIMilarity index (SSIM) of the two algorithms using the UDIS algorithm and the SuperUDIS algorithm. PSNR is the ratio of signal maximum power to signal noise power and is the most common and widely used objective measurement method for evaluating picture quality. The larger the value of PSNR between two images, the more similar the two images are. SSIM is an index to measure the similarity of two images. It measures the similarity between two images in terms of brightness, contrast, and structure. The value of the SSIM ranges from 0 to 1. A larger value indicates a smaller image distortion.

We calculate the average value of peak signal-to-noise ratio (PSNR) and structural similarity (SSIM) of the two methods on the SEAGULL dataset and the LPC dataset, respectively, and the specific results are shown in Table 1 and Table 2. It can be seen that the PSNR and SSIM of SuperUDIS on the two common datasets of image stitching are superior to AutoStitch [47] and the UDIS algorithm. The PSNR index increased 0.5 on average and the SSIM index increased 0.02 on average compared with the UDIS algorithm, indicating that the optimization made in this paper is conducive to the restoration of image stitching.

**Qualitative**: We compare the results of the proposed method with the previous method, including AutoStitch and the UDIS algorithm, as shown in Figure 14, Figure 15 and Figure 16.

Figure 14 shows the results using the AutoStitch, UDIS, and SuperUDIS algorithms on the SEAGULL dataset and the LPC dataset. In Figure 14a. represents the situation in which traditional methods can perform well but deep learning methods perform poorly; Figure 14b. represents the situation in which deep learning methods can perform well but traditional methods perform poorly; and Figure 14c. represents the situation in which both methods perform equally badly. However, SuperUDIS performs well in three situations. This proves that our method combines the advantages of traditional methods and deep learning methods and can complete the stitching work in a variety of complex environments.

Figure 15 and Figure 16 show the results of concatenated images on the SEAGULL dataset and the LPC dataset, respectively. The top two sets of images in Figure 15 and Figure 16 are stitched by the UDIS algorithm, and the bottom two sets of images are completed by the SuperUDIS algorithm. It can be seen that the red boxes in Figure 15a–d all have obvious dislocation phenomena, and the original straight line structure in Figure 15a,b yellow boxes has obvious distortion after the UDIS algorithm. In the four results obtained by using SuperUDIS, because we use the Laplace operator instead of the first difference in the original loss function and emphasize the continuity of the structural boundary, the dislocation phenomenon is obviously avoided. In addition, the use of Superpoint and Lightglue methods instead of the gridded unsupervised warping used in the original UDIS avoids distortion of the linear structure. In the results spliced by the UDIS algorithm in Figure 16a the runway in Figure 16b, the railings in Figure, and Figure 16d the steps in Figure, there are obvious dislocations; in the results of Figure 16c, there is also a distortion of the linear structure of the wall; Figure 16c,d There are obvious mutations on both sides of the joint, which seriously affect perception. In the results of the SuperUDIS algorithm, the dislocation stitching in the results of Figure 16a,b,d was repaired, and the distortion of the linear structure in Figure 16c was also repaired. Due to the chroma balance algorithm proposed in this paper, the color transition on both sides of the suture line in Figure 16c,d was significantly smoother, which greatly improved the perception of human eyes.

To sum up, whether qualitative or quantitative, it can be seen that the stitching results obtained by the model using our modified loss function are superior to the UDIS algorithm. Therefore, it can be concluded that the method in this paper can modify and improve the UDIS algorithm.

### 4.3. Ablation Study

In order to prove the optimization contribution of each part of the algorithm to the whole algorithm, an ablation experiment is carried out in this part, and each part of the algorithm is combined with the rest of the original algorithm, and quantitative and qualitative experiments are carried out.

**Quantitative:** The quantitative experimental results are shown in Table 3 and Table 4. The algorithm is divided into three parts: Superpoint + Lightglue, Chroma Balance, and Improved UDIS. The first row in the table indicates that only Superpoint and Lightglue algorithms are used: Superpoint and Lightglue algorithms are used in the image distortion part, and the UDIS fusion algorithm is used without an optimized loss function. The second line indicates that only the chroma balance algorithm is used: the unsupervised distortion algorithm of UDIS is used for the distorted part of the image, and then the color is corrected using the chroma balance algorithm, and finally the UDIS fusion algorithm without the optimized loss function. The third line represents the UDIS fusion algorithm using only an optimized loss function: the UDIS distortion algorithm is used in the image distortion part, and the UDIS fusion algorithm with an improved loss function is used in the image fusion part. The fourth line represents the simultaneous use of Superpoint, Lightglue, and Chroma Balance algorithms: in the distorted part of the image, Superpoint and Lightglue algorithms were used, followed by the Chroma Balance algorithm to correct the color, and finally the UDIS fusion algorithm without the optimized loss function. The fifth line indicates the simultaneous use of the three methods in this paper, namely the SuperUDIS algorithm proposed in this paper. The sixth line is the complete UDIS algorithm.

As can be seen from the results in the table, compared with the UDIS algorithm, PSNR and SSIM are improved when Superpoint + Lightglue is used alone, and UDIS is improved, while the Chroma Balance algorithm is mainly to optimize color continuity, with a slight decrease in PSNR but also an improvement in SSIM. The results of using the Superpoint, Lightglue, and Chroma Balance algorithms at the same time are also improved compared to using the two algorithms alone. The final SuperUDIS algorithm in this paper also has better PSNR and SSIM data than several algorithms alone. This shows that the three algorithms used in this paper are superior to the UDIS algorithm, complement each other, and can obtain better stitched results.

**Qualitative:** The qualitative experimental results are shown in Figure 17 and Figure 18.

In the example shown in Figure 17, the UDIS stitching result has one misalignment (red box) and one misalignment seam (yellow box). After Superpoint + Lightglue is used, the dislocation is obviously gone and the resolution is improved, but the original 4 × 4 grid becomes 2 × 4, indicating that there are problems in the joint planning. Since there is no large color difference between the two graphs in the example of Figure 17, there is no significant change in the balance using only the color difference. When the UDIS fusion algorithm with an improved loss function is used, it can be seen that the original dislocation and joint planning problems are optimized. In the final SuperUDIS algorithm proposed in this paper, it can be seen that not only the original two problems are solved, but also the resolution is improved.

In the example in Figure 18, the result of the UDIS stitching has a significant color difference (red box) and a distortion of the straight structure (yellow box). After using only Superpoint + Lightglue, the distortion of the linear structure was obviously repaired, but the color difference did not disappear. When only the chroma balance algorithm is used, the color difference is obviously weakened, but the distortion of the linear structure still exists. When the UDIS fusion algorithm with an improved loss function is used, it can be seen that the mesh-like floor tile connection is smoother, but the color difference and distorted straight line structure are not repaired. In the final use of the SuperUDIS algorithm, the distortion and color difference of the linear structure are eliminated, and the floor tiles are smoothly connected.

To sum up, the three improvements to the UDIS algorithm in this paper are all valuable and meaningful, and when the three algorithms work together, they can provide better stitching results.

## 5. Conclusions

In this paper, Superpoint feature point extraction, Lightglue feature point matching, and improved UDIS unsupervised fusion are combined for the first time, a chroma balance algorithm is added for optimization, and a more effective SuperUDIS image stitching method is obtained.

First of all, this paper uses Superpoint and Lightglue, which extract feature points efficiently and accurately, and the UDIS unsupervised stitching method, which can generate floating point masks, in the image stitching part to make image stitching smoother. In addition, this paper optimizes the existing problems of the UDIS method. Aiming at UDIS color mutation in the transition part of the image, this paper designs a chroma balance algorithm to make the color of the transition part of the image smoother. In order to emphasize the continuity of the structure edge during training, the loss function of UDIS is optimized by replacing the difference of horizontal and vertical directions with the Laplacian operator of second-order differentiation. Compared with the previous UDIS algorithms, our method inherits the advantages of the previous algorithms, optimizes the problems of the UDIS algorithm, and improves the PSNR and SSIM. SuperUDIS can give better stitching results for both conventional scenes and scenes with a large color difference or multi-linear structure.

Moreover, experiments on various datasets show that the proposed SuperUDIS method outperforms existing image stitching methods in terms of image quality and stitching accuracy. The chroma balance algorithm effectively reduces color mutations in the transition part of the image, resulting in a smoother and more visually appealing stitched image. The improved UDIS loss function enhances the continuity of structure edges in the stitched image, leading to better overall stitching results.

In conclusion, the combination of Superpoint feature point extraction, Lightglue feature point matching, and improved UDIS unsupervised fusion, along with the addition of the chroma balance algorithm, results in a more effective and superior image stitching method known as SuperUDIS. This method has shown promising results in various experiments and is capable of handling different types of scenes with different levels of complexity. This research contributes to advancing the field of image stitching and paves the way for further improvements in image processing algorithms.

## Figures and Tables

**Figure 1 sensors-24-05352-f001:**
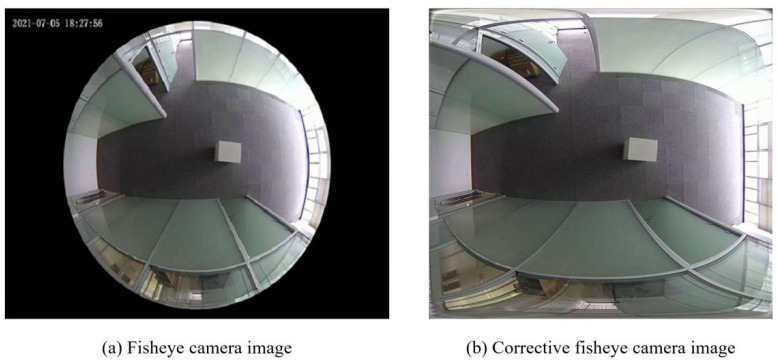
Fisheye camera image and corrective fisheye camera image. The fisheye lens has distortion; even after post-processing the image, edge information will still be lost.

**Figure 2 sensors-24-05352-f002:**
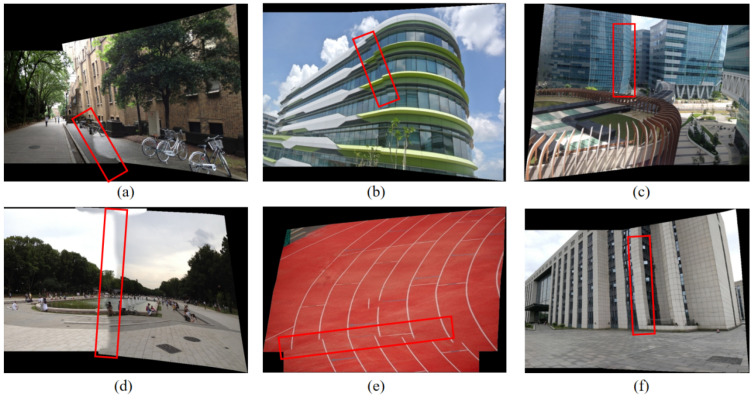
Problems in the UDIS ++ algorithm. The rectangle highlights the areas with different type of poor stitching. (**a**,**d**) Discontinuity of color transitions; (**b**,**e**) Poor stitching seam; (**c**,**f**) Linear structure distortion.

**Figure 3 sensors-24-05352-f003:**
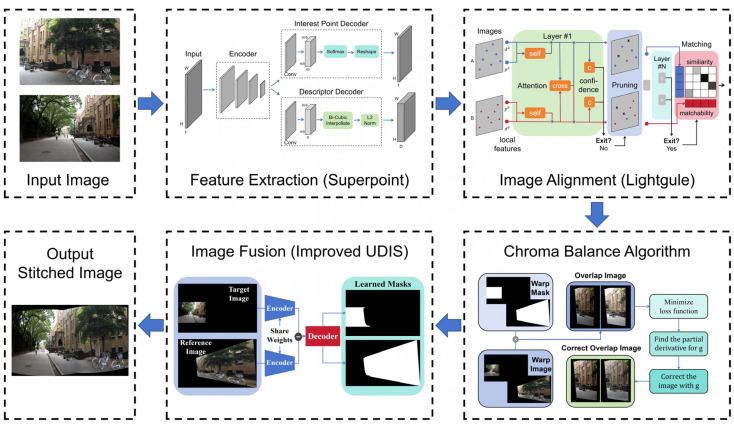
The Architecture of SuperUDIS. The algorithm consists of four parts: feature extraction, image alignment, chroma balance, and image fusion.

**Figure 4 sensors-24-05352-f004:**
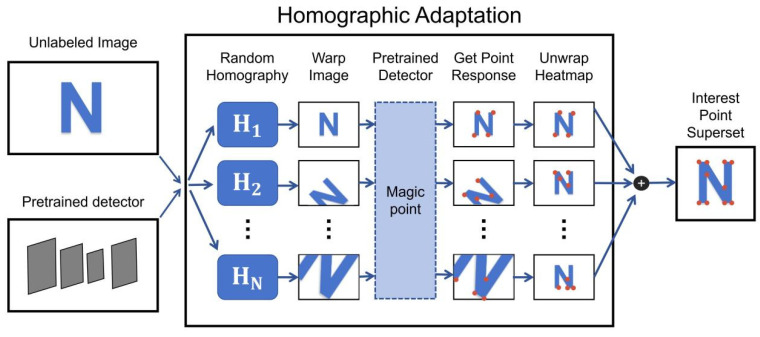
The Architecture of Homographic Adaptation. Homographic adaptation uses different affine transformations to get different interest points on the initial image and empirically sums up a large enough number of random samples to get a more adaptable interest point detector.

**Figure 5 sensors-24-05352-f005:**
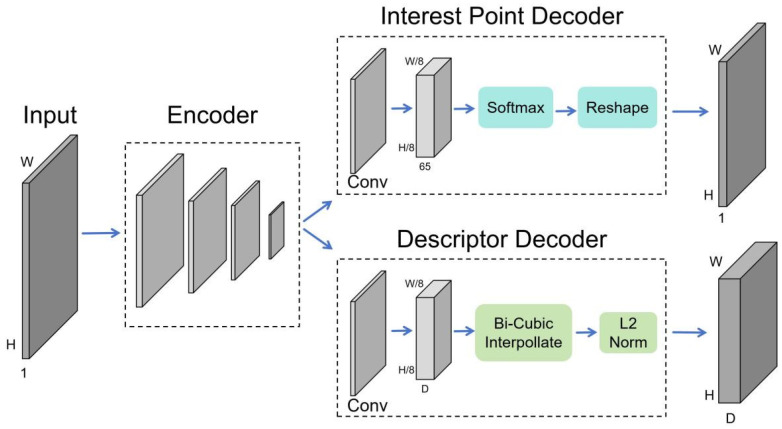
The Architecture of a Superpoint Decoder. The operation calculation is carried out by the interest point decoder and descriptor decoder at the same time, which improves the operation efficiency.

**Figure 6 sensors-24-05352-f006:**
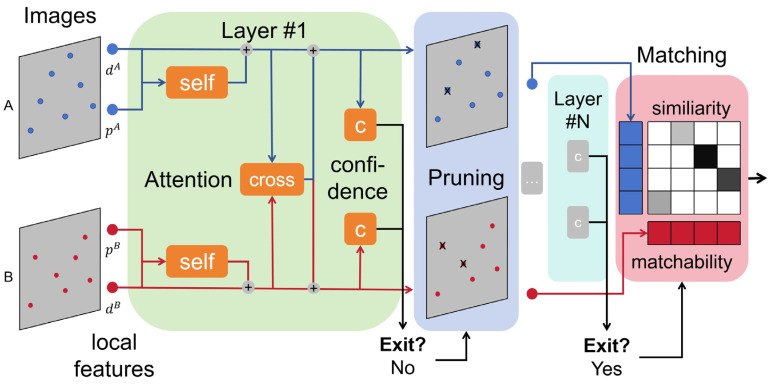
Lightglue Architecture. Given a set of input image feature points (d,p), each layer uses a self-attention unit and a cross-attention unit to update the state of the feature points. Then, a confidence classifier c helps to decide whether to stop the inference. Additionally, the confidently unmatchable point will be pruned. Finally, a lightweight head calculates the match.

**Figure 7 sensors-24-05352-f007:**
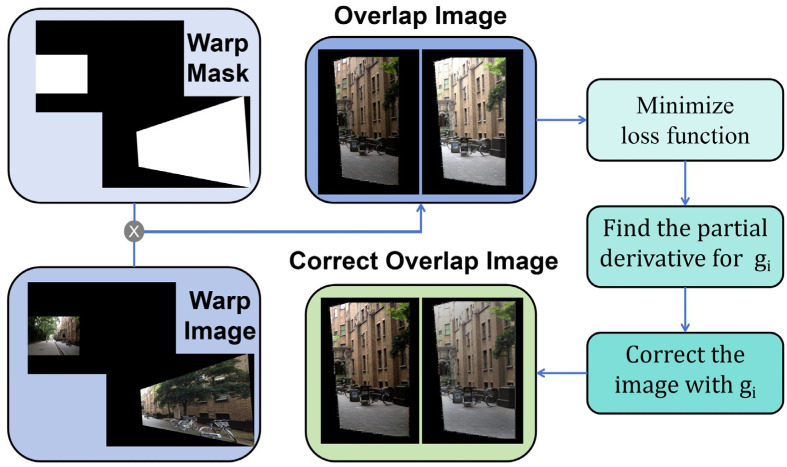
Chroma Balance Algorithm. The main purpose of the chroma balance algorithm is to balance the chroma difference between the overlapping parts of two images and ensure the smoothness of the colors on both sides of the suture line after fusion.

**Figure 8 sensors-24-05352-f008:**
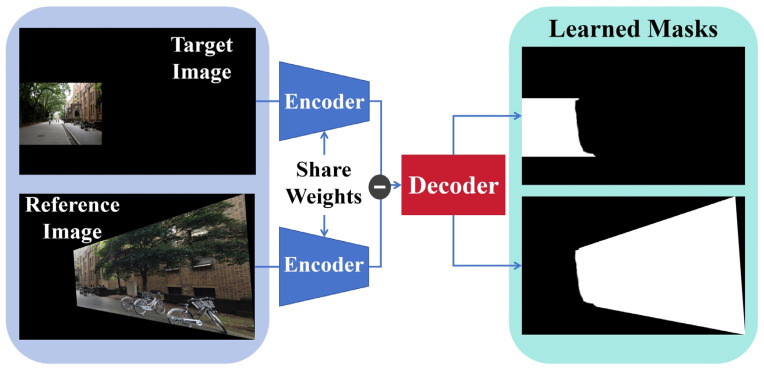
UDIS Fusion Algorithm. The algorithm can generate a floating mask according to the trained model, which makes the final result smoother.

**Figure 9 sensors-24-05352-f009:**
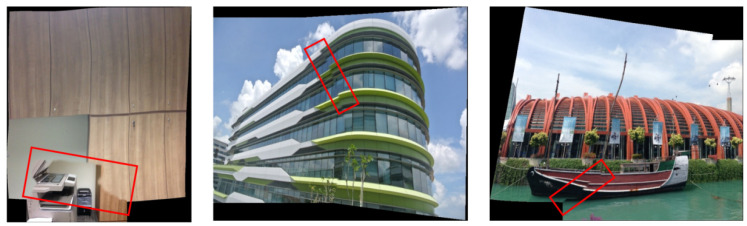
Poor Stitching Seam by UDIS++. The rectangle highlights the areas with misalignment and blurring in some images realized by UDIS++.

**Figure 10 sensors-24-05352-f010:**
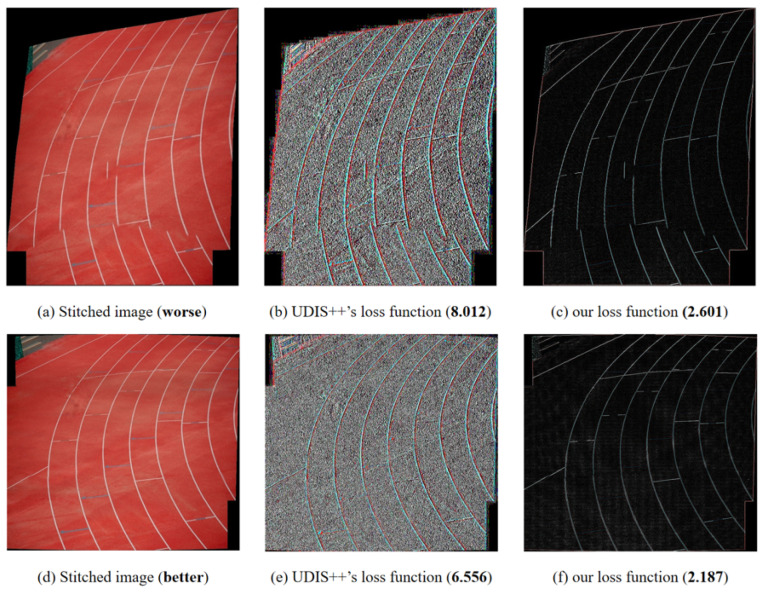
Loss Function Effect. (**a**,**d**) are the source stitched images; (**b**,**e**) are the results operated by the horizontal difference of the UDIS++ loss function; and (**c**,**f**) are the results operated by our loss function.

**Figure 11 sensors-24-05352-f011:**
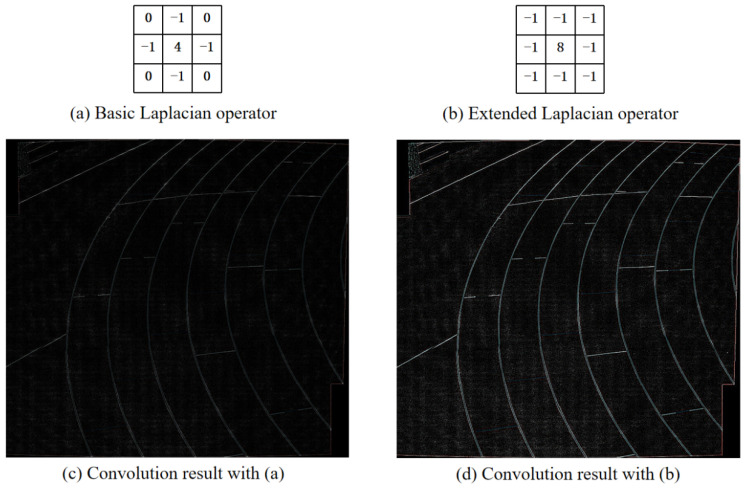
Laplacian operator. (**c**,**d**) are the convolution results with (**a**,**b**).

**Figure 12 sensors-24-05352-f012:**
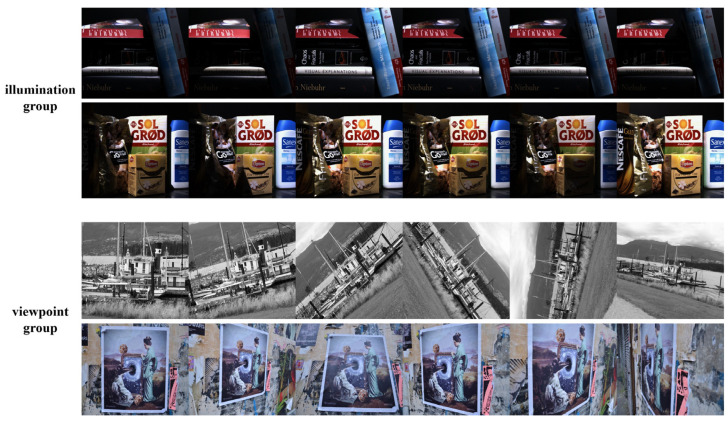
Part of the HPatches dataset. The dataset consists of an illumination group and a viewpoint group, with the upper two lines as examples of the illumination group and the lower two lines as examples of the viewpoint group.

**Figure 13 sensors-24-05352-f013:**
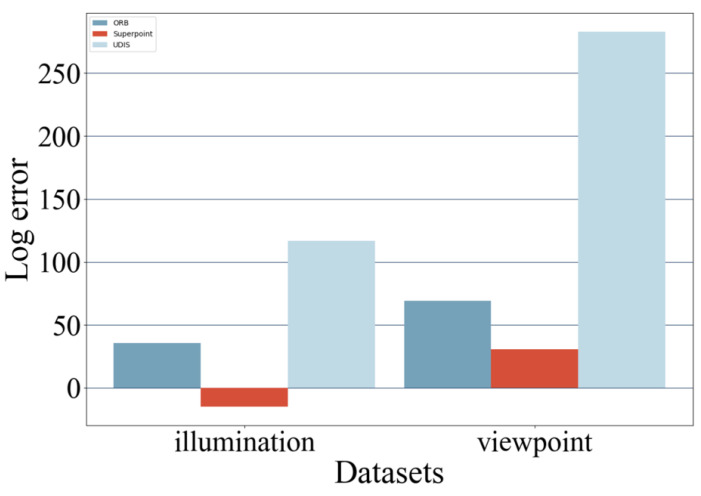
Image warp experiment result. The errors in H-matrix estimation by the three algorithms under different illumination and different viewpoints (after logarithm).

**Figure 14 sensors-24-05352-f014:**
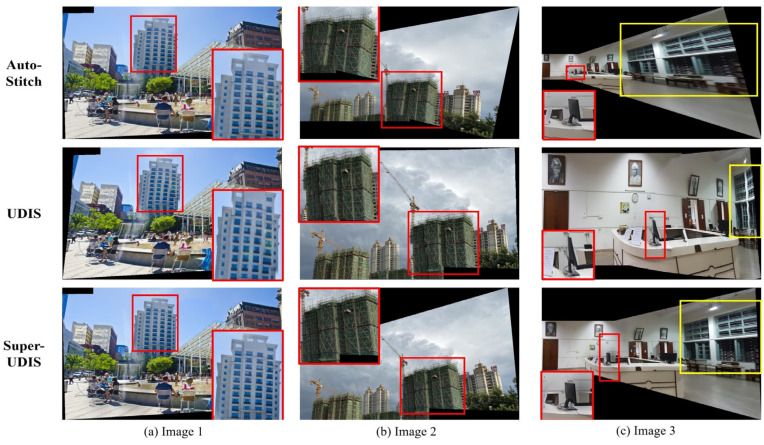
Comparison of the results using different methods on the LPC and SEAGULL datasets. The rectangle highlights the areas with stitching dislocation in the previous method, while our method can complete the stitching without misalignment or other problems.

**Figure 15 sensors-24-05352-f015:**
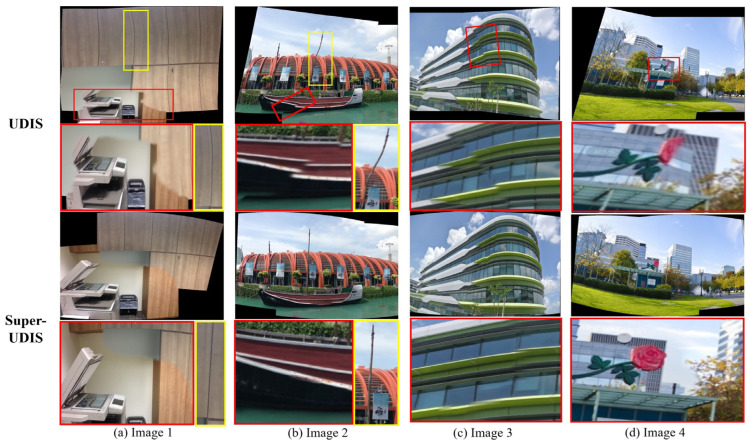
Comparison of the results using UDIS and SuperUDIS on the SEAGULL dataset. The red and yellow rectangle highlights the areas with stitching dislocation and linear structure distortion.

**Figure 16 sensors-24-05352-f016:**
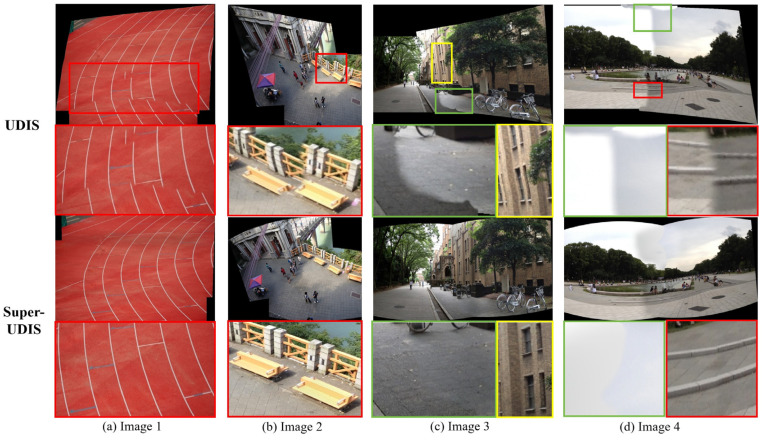
Comparison of the results using UDIS and SuperUDIS on the LPC dataset. The red rectangle highlights the areas with stitching dislocation, the green one highlights the areas with color discontinuity and the yellow one highlights the areas with linear structure distortion.

**Figure 17 sensors-24-05352-f017:**
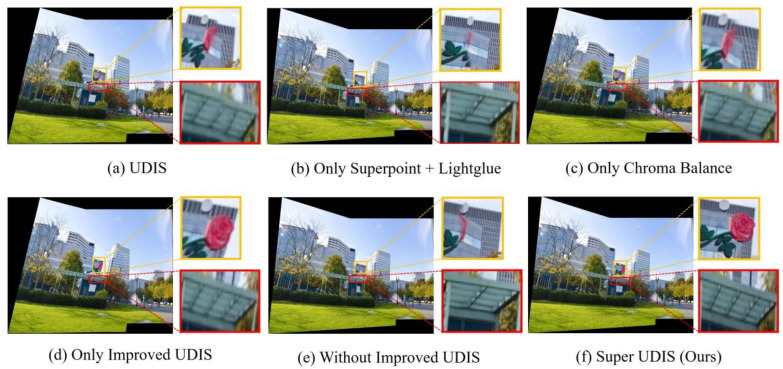
Ablation experiment results are compared on the SEAGULL dataset. The red and yellow rectangle highlights the areas with stitching dislocation.

**Figure 18 sensors-24-05352-f018:**
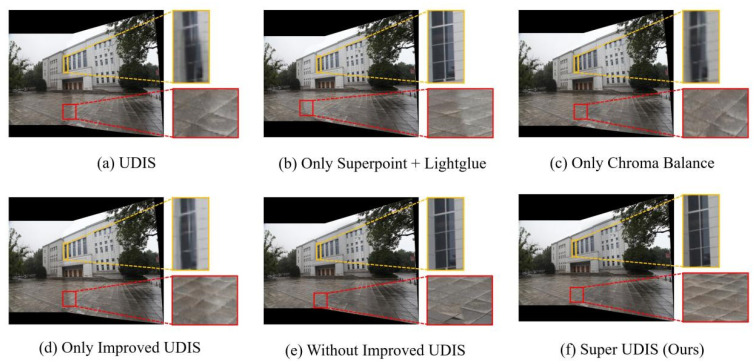
Ablation experiment results are compared to the LPC dataset. The red and yellow rectangle highlights the areas with color discontinuity and stitching dislocation.

**Table 1 sensors-24-05352-t001:** PSNR and SSIM on the SEAGULL dataset.

	AutoStitch	UDIS	SuperUDIS
PSNR↑	9.999	12.980	13.495
SSIM↑	0.4378	0.4362	0.4545

**Table 2 sensors-24-05352-t002:** PSNR and SSIM on the LPC dataset.

	AutoStitch	UDIS	SuperUDIS
PSNR↑	9.947	10.091	10.200
SSIM↑	0.4174	0.4424	0.4430

**Table 3 sensors-24-05352-t003:** PSNR and SSIM for ablation experiments on the SEAGULL dataset.

Method			PSNR	SSIM
Superpoint and Lightglue	Chroma Balance	Improved UDIS		
√			13.291	0.4417
	√		12.960	0.4399
		√	13.044	0.4426
√	√		13.475	0.4541
√	√	√	**13.495**	**0.4545**

**Table 4 sensors-24-05352-t004:** PSNR and SSIM for ablation experiments on the LPC dataset.

Method			PSNR	SSIM
Superpoint and Lightglue	Chroma Balance	Improved UDIS		
√			9.991	0.4294
	√		10.060	0.4365
		√	10.086	0.4423
√	√		10.054	0.4383
√	√	√	**10.200**	**0.4430**

## Data Availability

Data publicly available: The SEAGULL dataset is available at https://doi.org/10.1007/978-3-319-46487-9_23, accessed on 17 September 2016. The LPC dataset is available at https://github.com/dut-media-lab/Image-Stitching, accessed on 2021. The UDIS source codes is available at https://github.com/nie-lang/UDIS2, accessed on 22 July 2023. The Lightglue source codes is available at https://github.com/cvg/LightGlue, accessed on 23 June 2023. The data that supports the findings of this study is available from the corresponding author upon reasonable request.

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
