# Peer review of "Improved Unsupervised Stitching Algorithm for Multiple Environments SuperUDIS"

_sensors, 2024, doi:10.3390/s24165352_

Round 1

Reviewer 1 Report

Comments and Suggestions for Authors

The authors propose an improved image stitching algorithm called SuperUDIS. This algorithm combines the feature extraction and matching capabilities of Superpoint and Lightglue with the UDIS algorithm, while also integrating a chromaticity balance algorithm and an optimized loss function. SuperUDIS addresses issues present in previous methods, such as linear structure distortion, low resolution, and unnatural color transitions, performing exceptionally well in complex environments. Experimental results show that this method improves both PSNR and SSIM metrics. This research provides new insights for the development of image stitching technology and has the potential to play a significant role in various application scenarios.

1. In Figure 12, the authors present a comparison of H-matrix estimation errors for three algorithms under different illumination conditions and viewpoints. It is recommended that the authors clarify the specific illumination values and viewpoint values in this section. For example, they could specify the lighting conditions such as dusk, dawn, and night.

2. The authors should clearly explain in the section discussing Figure 12 why Superpoint + Lightglue results in the smallest error. They should provide a detailed analysis of the factors contributing to this superior performance.

3. The number of comparative algorithms in Table 1 and Table 2 is insufficient. It is recommended to add several more state-of-the-art algorithms.

4. It is recommended that the authors add visualization results of several state-of-the-art algorithms in Figures 13 and 14.

5. In the previous sections, the authors used both SEAGULL and LPC datasets for experiments. However, Table 3 only shows the ablation study on the SEAGULL dataset. It is recommended that the authors also include the ablation study on the LPC dataset.

Comments on the Quality of English Language

no

Reviewer 2 Report

Comments and Suggestions for Authors

The overall quality of the proposed work seems quite high. The method is robust and well-explained. The methodology is tested on various datasets, including the HPatches dataset, and comparisons are made with existing algorithms like ORB and UDIS. This comprehensive testing helps validate the robustness of the proposed method. However, some issues have been detected and the comments about them are provided below:

In Section 4.2, the authors propose a comparison with the state of the art. However, I only noticed results obtained on some datasets. Where is the comparison?

There are numerous typos and errors everywhere that should be fixed. (e.g., "Comparion to the State of the Art Methods"). Please, fully revise the manuscript, it is almost unreadable. There are also figures with grammar errors...

Minor:

Figure 3, some elements are unreadable, such as some variables (in Chrome Balance Algorithm, g of i?). Also, "confide-nce", as in Figure 6, has a newline in the middle of the word. Please, fix it.

The references could be updated thus the most recent is from 2020. I suggest an integration of the section. Moreover, the bibliography structure is not standardized; please, fix it.

Comments on the Quality of English Language

The English should be completely revised.

Round 2

Reviewer 2 Report

Comments and Suggestions for Authors

The authors addressed all the highlighted comments.

The paper is now eligible for its publication.

Comments on the Quality of English Language

The English is still not perfect, but (at least) there are no typos anymore.